# Technical note: Statistical generation of climate-perturbed flow duration curves

Veysel Yildiz[1], Robert Milton[2], Solomon Brown[2], and Charles Rougé[1]

[1]Department of Civil and Structural Engineering, The University of Sheffield, Sheffield, United Kingdom of Great Britain
[2]Department of Chemical and Biological Engineering, The University of Sheffield, Sheffield, United Kingdom of Great Britain

**Correspondence:** Veysel Yildiz (vyildiz1@sheffield.ac.uk)

**Abstract.**

Assessing the robustness of a water resource system's performance under climate change involves exploring a wide range of streamflow conditions. This is often achieved through rainfall-runoff models, but these are commonly validated under historical conditions with no guarantee that calibrated parameters would still be valid in a different climate. In this note, we introduce a new method for the statistical generation of plausible streamflow futures. It flexibly combines changes in average flows with changes in the frequency and magnitude of high and low flows. It relies on a three-parameter analytical representation of the flow duration curve (FDC) that has been proved to perform well across a range of basins in different climates. We rigorously prove that for common sets of streamflow statistics mirroring average behavior, variability, and low flows, the parameterisation of the FDC under this representation is unique. We also show that conditions on these statistics for a solution to exist are commonly met in practice. These analytical results imply that streamflow futures can be explored by sampling wide ranges of three key flow statistics, and by deriving the corresponding FDC to model basin response across the full spectrum of flow conditions. We illustrate this method by exploring in which hydro-climatic futures a proposed run-of-river hydropower plant in eastern Turkey is financially viable. Results show that contrary to approaches that modify streamflow statistics using multipliers applied uniformly throughout a time series, our approach seamlessly represents a large range of futures with increased frequencies of both high and low flows. This matches expected impacts of climate change in the region, and supports analyses of the financial robustness of the proposed infrastructure to climate change. We conclude by highlighting how refinements to the approach could further support rigorous explorations of hydro-climatic futures without the help of rainfall-runoff models.

## 1 Introduction

Projections of climate change and its impact on water resources are inherently uncertain, and this is likely to increase as a result of climatic, technological, economic and sociopolitical changes (Maier et al., 2016; McPhail et al., 2018). Water resource planners and decision makers are rightly concerned about the potential effects of future uncertainties, with the upfront cost of action to be weighed against the high potential social and environmental costs of inaction over time (Singh, 2018; Ray et al., 2018). Conventional engineering approaches to water systems planning have been summarised as "predict-then-act" (Lempert

et al., 2013), with optimisation of a design objective under the assumption of a best-estimate (i.e., most likely) prediction of the future suggesting the "best" course of action. To produce future streamflow in this framework, rainfall-runoff models are routinely forced by rainfall and temperature projections of dynamically downscaled global climate models (GCMs; Peel and Blöschl, 2011; Chen et al., 2019). There are, however, two categories of issues with this type of approach.

First, "predict-then-act" is not compatible with hard-to-quantify uncertainties, as it works best when a known single probability density function is available for each key parameter (Singh et al., 2015). If the future turns out to be different from the hypothesized projection(s), the optimal solution could fail, sometimes catastrophically (Haasnoot et al., 2013; Hamarat et al., 2013). To avoid this, several emerging decision-making frameworks (Lempert, 2002; Bryant and Lempert, 2010; Brown et al., 2012; Haasnoot et al., 2013; Kasprzyk et al., 2013) strive to find adaptation solutions that are robust to uncertain and changing conditions. In the climate adaptation context, a robust alternative maintains satisfactory expected performance under a range of plausible futures (Maier et al., 2016; McPhail et al., 2018; Marchau et al., 2019), instead of being "optimal" in a single future. Therefore, to identify robust alternatives, uncertainties have to be described with the aid of scenarios that represent coherent future pathways based on different sets of assumptions (Maier et al., 2016). In water resource applications, this entails defining specific ranges for future uncertainties including streamflow, then sampling them to generate an ensemble of plausible future conditions.

The second category of issues is with the use of rainfall-runoff models to generate future flow conditions. Indeed, these models have generally been calibrated and validated under historical conditions, with no assurance that these parameters would still be valid under different hydro-climatic conditions (Peel and Blöschl, 2011). There is evidence that rainfall-runoff models' predictive skill decreases with changed climatic conditions (Saft et al., 2016; Seibert et al., 2016; Fowler et al., 2020). In fact, a study of the Rhine-Meuse basin from 1901 to 2010 shows that optimal calibration evolves with climate variability, and land use and river structure change (Ruijsch et al., 2021). To compound these calibration issues, the significant resources and modelling skill needed for calibration and validation mean that is costly for water resource assessments based on rainfall-runoff models to explore the full uncertainty space associated with climate change, with far-reaching consequences for planning.

For these reasons, approaches aimed at finding climate-robust adaptation solutions have often relied on multipliers applied uniformly along a time series also known as the "delta change" approach (Brown et al., 2012). Examples of this affect streamflow either directly through multiplication (e.g., Herman et al., 2014, 2015) or indirectly by applying to climate variables such as temperature and precipitation, before using regression to deduce annual runoff (e.g. Ray et al., 2018). More sophisticated versions of this exist, e.g., Quinn et al. (2018) distinguished several multipliers to isolate changes to the mean, to variance, and to monsoonal dynamics in the Red River basin in Vietnam. However, to our knowledge there is no approach that seeks to describe catchment response under changing climate in a coherent way across the full range of hydrological conditions.

As the representation of the empirical cumulative distribution function (CDF) of streamflow (Vogel and Fennessey, 1994), a flow duration curve (FDC) precisely represents the full range of hydrological conditions. The FDC is unique to each catchment, and it is influenced by various factors including climate, topography, physiography, vegetation cover, land use (Castellarin et al., 2013; Brown et al., 2013; Sadegh et al., 2016). It has become a popular tool used in modern hydrology for various water resources applications (Leong and Yokoo, 2021), since it provides concise and valuable information about river streamflow

variability and catchment response (Blöschl et al., 2013; Boscarello et al., 2016). For example, slope steepness in the middle part of a FDC is characteristic of a catchment's precipitation retention properties (Yilmaz et al., 2008).

This remark has led Sadegh et al. (2016) to adapt a set of soil retention functions such as those proposed by van Genuchten (1980) and Kosugi (1996) to mimic the empirical FDCs of catchments. These models are used in soil physics and hydrology to characterise water flow in unsaturated soils and to estimate soil water retention properties. This analogy is based on the idea that both watersheds and soils are governed by similar hydroclimatologic forcing, and are able to store and dispel precipitation in response to similar gradients (Vrugt and Sadegh, 2013; Sadegh et al., 2016). Fitting FDCs to a set of 430 catchments of the MOPEX dataset (Duan et al., 2006), Sadegh et al. (2016) found that the three-parameter Kosugi model they proposed offered the best quality of fit across a broad range of climate zones, under a goodness-of-fit criterion that weighs high and low flows equally. It is based on a lognormal distribution with three parameters (Kosugi, 1994, 1996) that are determined by calibration against the empirical FDC of a watershed.

This paper leverages the existence of high-performing parameterisations of the FDC across a range of climates to statistically generate plausible streamflow futures. We directly link parameter triplets of the Kosugi model with three streamflow statistics that are relevant to the management of water resources: central tendency, variability, and low-flow indicator. This one-on-one correspondence enables us to (1) sample hydro-climatic futures according to plausible ranges for streamflows statistics, and (2) convert these into ensembles of FDCs that represent the differentiated impacts of climate change across flow quantiles. The latter is consistent with studies of historically observed streamflow change (e.g. Pumo et al., 2016).

## 2 Methodology

This section demonstrates the technique that is the core of this paper, and introduces its workflow. First, Section 2.1 will introduce the Kosugi model of the flow duration curve (FDC). Then Section 2.2 will give results on how to parameterise the FDC with the Kosugi model to reproduce desired streamflow statistics. These are the key results that enable us to build the methodological workflow to produce an ensemble of climate-perturbed flow duration curves, which we present in Section 2.3.

### 2.1 Kosugi model of the flow duration curve

The flow duration curve (FDC) is a cumulative frequency curve that ranks the observed record of $n$ discharge values in descending order $\{q_1, q_2, \ldots, q_n\}$. The ranking of each value directly gives its empirical probability of exceedance $u$. In this work, we represent the FDC with the three-parameter Kosugi model, which has been shown to provide an excellent approximation to FDCs under a wide range of climates (Sadegh et al., 2016), and is given by:

$$q(u) = c + (a - c)\, z(u)^b, \ \text{with} \ z(u) = \exp\left[\sqrt{2}\, \text{erfc}^{-1}(2\,u)\right] \tag{1}$$

where $q$ is the streamflow value for a given value of the exceedance probability $u \in [0, 1]$, $(a, b, c)$ are the three coefficients of the Kosugi model, and erfc is the complementary error function. Given a discharge record, the Kosugi model is fitted by minimising the root mean square srror (RMSE). Minimising the RMSE on $q(u)$ would lead to weigh errors in the high flows

more than those on the low flows. For this reason, we minimise the RMSE in the exceedance probability space, i.e., the error on $\mathcal{U}$, the inverse of the $q(u)$ function defined in equation (1):

$$RMSE(x) = \left[ \frac{1}{n} \sum_{i=1}^{n} [u_i - \mathcal{U}(q_i|a,b,c)]^2 \right]^{0.5} \quad \text{where} \quad \mathcal{U}(q|a,b,c) = \frac{1}{2} \, \text{erfc} \left[ \frac{1}{\sqrt{2}\,b} \ln \left( \frac{q-c}{a-c} \right) \right] \quad (2)$$

To fit the Kosugi model and capture flow variability within the FDC, it is necessary to have daily discharge measurements over a sufficient period of time, e.g., more than 20 years.

## 2.2 Correspondence between common flow statistics and the Kosugi model

In this paragraph, we directly relate the three parameters of the Kosugi FDC model with sets of three streamflow statistics that are of interest to water resource management. This is key to relating a hydro-climatic future (described with different flow statistics) to a well-defined FDC. The central tendency, and the spread or the degree of variation are the two key aspects to describing a distribution (Weisberg and Weisberg, 1992; McCluskey and Lalkhen, 2007). Low flows are also of interests where water scarcity and availability are issues. With this we construct a triplet of streamflow statistics $(M, V, L)$ where $M$ is the central tendency (mean or median), $V$ is variability (standard deviation or coefficient of variation), and $L$ can be given by a low flow quantile (first or fifth percentile of flow distribution).

We can entirely define the flow distribution associated to a hydro-climatic future defined by $(M, V, L)$, if we can find a relationship relating it to parameters $(a, b, c)$ of the Kosugi model defined in equation (1):

$$(a, b, c) = \mathcal{F}(M, V, L) \quad (3)$$

This correspondence needs to be unique: if there is more than one $(a, b, c)$ for a future defined by $(M, V, L)$, a method based on the Kosugi model cannot define future flows unambiguously. In this paper we focus on two sets of $(M, V, L)$. On the one hand, using $M$ as the mean, $V$ as the standard deviation and $L$ as a low flow percentile corresponds to a very common statistical description of a flow distribution. We will refer to this as the "mean" case hereafter. On the other hand, there are cases where using the median, coefficient of variation and low flow quantile as $(M, V, L)$ is of interest. This is the case e.g., in appraisals of run-of-river hydropower, see Section 3. We will refer to this as the "median" case hereafter.

Step by step derivation of these equations, along with proof of the uniqueness of a parameterisation, and conditions on the existence of solutions are provided in the Supplementary Information (SI) to this paper. In this section, we provide the main results for both the "mean" and "median" cases.

### 2.2.1 "Mean" case

In the "mean case", we know $(M, V, L) = (\mu, \sigma, q_{low})$ where where $\mu$ is the mean, $\sigma$ is the standard deviation and $q_{low}$ is the $1^{st}$ or $5^{th}$ percentile of flow. To parameterise the Kosugi equation in this case, one needs to first find $b$ that is solution of:

$$\frac{\sigma}{\mu - q_{low}} = \frac{\sqrt{e^{b^2} - 1}}{1 - e^{-b^2/2} \varepsilon^b} \quad (4)$$

where $\varepsilon$ is the value of $z(u)$ at $q_{low}$. For instance $\varepsilon = z(0.99) \approx 0.0976$ if $q_{low}$ is the first percentile, and $\varepsilon = z(0.95) \approx 0.1930$ if $q_{low}$ is the fifth percentile. There is at most one solution to this equation, and it exists if:

$$\frac{\sigma}{\mu - q_{low}} > \frac{-1}{\ln(\varepsilon)} \tag{5}$$

where $\varepsilon < 1$ so $\ln(\varepsilon) < 0$ and $-1/\ln(\varepsilon) \approx 0.43$ if $q_{low}$ is the first percentile; $0.61$ if $q_{low}$ is the fifth percentile. Then one can deduce $a$ and $c$ using the following equations:

$$\begin{cases} a = \dfrac{q_{low} \left(1 - e^{-b^2/2}\right) + \mu\, e^{-b^2/2} (1 - \varepsilon^b)}{1 - e^{-b^2/2} \varepsilon^b} \\[2em] c = \dfrac{q_{low} - \mu\, e^{-b^2/2}\, \varepsilon^b}{1 - e^{-b^2/2} \varepsilon^b} \end{cases} \tag{6}$$

### 2.2.2 "Median" case

In the "median" case, we know $(M, V, L) = (m, CV, q_{low})$, where $m$ is the median, $CV = \mu/\sigma$ is the coefficient of variation, and $q_{low}$ continues being a low flow percentile. One parameter of the Kosugi equation is easy to obtain:

$$a = m \tag{7}$$

To find the other parameters it is necessary to find the $b$ that is the solution of:

$$CV = (1 - R) \frac{\sqrt{e^{b^2} - 1}}{1 - R + (R - \varepsilon^b)e^{-b^2/2}} \tag{8}$$

where $R = q_{low}/m$. $b$ is unique, and exists provided a similar existence condition as in the "mean" case:

$$\frac{CV}{1 - R} > \frac{-1}{\ln(\varepsilon)} \tag{9}$$

Then the final parameter $c$ is obtained through:

$$c = \frac{q_{low} - m\varepsilon^b}{1 - \varepsilon^b} \tag{10}$$

### 2.2.3 Domain of validity of existence conditions

In this paragraph, we explain what the conditions for the existence and uniqueness provided imply – see equations 5 and 9 for "mean" case and "median" case respectively. Both equations are equivalent to:

$$CV > \frac{-(1 - R)}{\ln(\varepsilon)} \tag{11}$$

where $0 < R < 1$ is a ratio of the low flows by the mean or median; recall that $-1/\ln(\varepsilon) \approx 0.43$ if the low flow parameter is the first percentile, or $0.61$ if it is the fifth percentile.

From equation 11, it is sufficient to have $CV > -1/\ln(\varepsilon)$ for both existence conditions to be verified. This condition has been verified for a large majority of the catchments over a large dataset of 6807 gages in the continental US (see Ye et al.

(2021)). Yet for the existence condition to not be met the multiplier of $(1 - R)$ must also be close to 1. In other words, low
flows must be extremely low relative to the mean (for the "mean" case) or median (for the "median" case), but this may be
incompatible with a low value of CV. In fact, in Figure 10 from Ye et al. (2021), all time series with zero flow days in the sample
have a CV value close or equal to 1. Together, these remarks suggest that the existence condition should be realised in most
cases where flows are not strongly regulated. However, we would like to point out that whether the conditions of equations 5
or 9 are met for historical flows is of limited relevance. They need to be verified for each plausible future flow for which a FDC
is generated. For this reason, we consider that checking these conditions across large databases of historical flows would be of
limited interest within the scope of this work.

## 2.3 Producing an ensemble of climate-perturbed flow duration curves

Figure 1 illustrates our four-step methodology. In step (1), we fit the Kosugi FDC model to the available discharge record
by finding the parameters $(a_h, b_h, c_h)$ for the historical record, using equation (3) and the chosen historical flow statistics
$(M_h, V_h, L_h)$. We need to verify that this fit is close in performance to the best-fit model $(a^*, b^*, c^*)$ obtained through RMSE
minimisation as described by equation (2). It is essential to prove that the FDC model provides a good representation of
historical observations, otherwise a perturbation of the model would be a poor representation of a perturbation of the historical
flow regime. We then check the method by deriving the FDC parameters based on three key statistics of historical flow. The
method can be used if both curves adequately fit the functional shape of the empirical FDC.

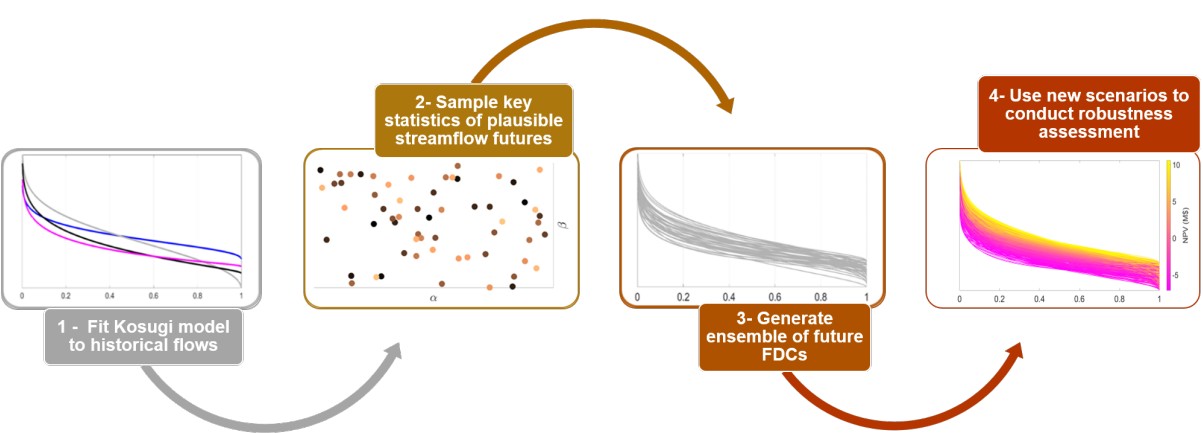

**Figure 1.** Flowchart of the approach; (1) Kosugi model parameters are calibrated with a historical FDC, (2) a set of scenarios with modified
flow statistics are determined, (3) a new set of Kosugi model coefficients are derived for each future scenario, and future scenarios are created
by using these coefficients, (4) future scenarios can be used in robustness assessments.

To generate future flows, one needs to sample a set of futures in step (2). This corresponds to sampling the chosen parameters
$(M, V, L)$ to construct an ensemble $\{(M_i, V_i, L_i)_{1 \leq i \leq N}\}$ of $N$ alternative futures, reflecting a broad range of plausible future

conditions. Then in step (3), we find the unique set of parameters $(a_i, b_i, c_i)$ for each triplet $(M_i, V_i, L_i)$ and construct the corresponding FDC. Finally in step (4), we use the resulting ensemble of FDCs to support robustness assessments in a changing climate, by evaluating the performance of a decision adaptation(s) across future scenarios.

Note that the first three steps of this workflow can be replicated for any site using the Zenodo repository (Yildiz et al., 2022b) that accompanies this paper. The fourth step depends on the specificities of each robustness assessment, e.g., what infrastructure is considered, what performance measures, etc.

## 3 Case study

This section demonstrates the fitness of our method for robustness assessments.

### 3.1 Site description

The case study involves the climate change impact analysis of a proposed RoR hydropower plant at the Besik site on the Mukus River in Van province located in the Eastern Anatolia region of Turkey (Lat: 38.15°N; Lon: 42.80°E). Summers are dry and hot with temperatures above 30 °C. Spring and autumn are generally mild, but during both seasons sudden hot and cold spells frequently occur. 27 years of daily discharge observation are available. The discharge fluctuates considerably between values of 2 and 38 m³/s, with median flow of 4.79 m³/s, first percentile flow of 2.23 m³/s and coefficient of variation of 0.60. The design of the run-of-river hydropower project was optimised using the HYPER toolbox (Yildiz and Vrugt, 2019). The resulting design has an installed capacity of 8.73 MW, a penstock length of 208 m with a diameter of 1.60 m, and two side-by-side Francis turbines whose design discharge are 4.80 and 2.87 m³/s respectively.

### 3.2 Generation of climate-perturbed flow duration curves

Contrary to reservoir-based hydropower plants, RoR schemes have virtually no storage, so they are vulnerable to changes in flow as they cannot modulate flows and only operate in a predefined range. Extreme low flows are insufficient to activate the turbines, and equally, flows above the design discharge do not produce additional energy. Because of this focus on the mid-range flows, the median is a more important indicator of performance than the mean flow, which can be skewed by high discharges. For this reason, this application will relate median, coefficient of variation and first percentile flows to Kosugi parameters (the "median" case).

In step (1) of our approach, we fit the three-parameter Kosugi model to the daily discharge data. Figure 2 shows the historical records (red circles), the fitted Kosugi Model (black line) and the derived FDC based on the three statistical parameters of historical records (FDC from $(M_h, V_h, L_h)$). Both fitted curves offer close fits across the entire spectrum of flow conditions described by the FDC of historical records. In particular, the quality of the fit for middle and low flows shows the consistency of the proposed approach, as their estimation is vital in assessing and managing water resources such as hydropower plants.

In step (2), we determine plausible ranges for the three statistical parameters over the operational life of the proposed plant. In Turkey, hydropower projects are licensed to generate electricity for a period of 49 years. Several climate projections

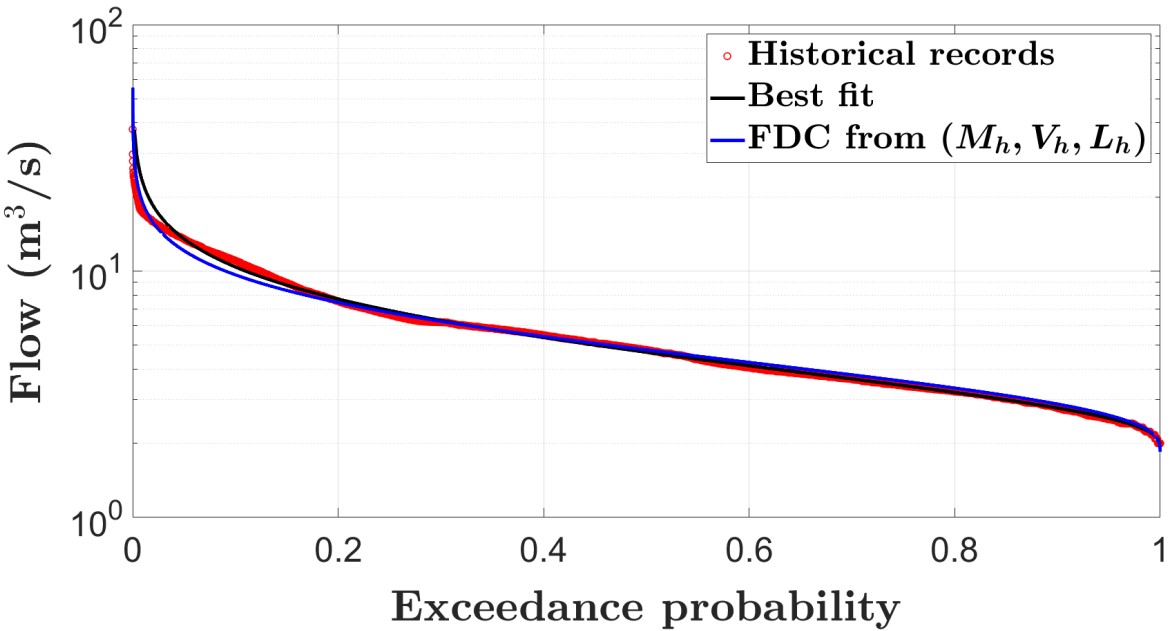

**Figure 2.** Plot of the daily flow duration curves (FDC) used in the case study (red circles). Black line represents the fitted Kosugi model and the blue line is the FDC deduced from $(M_h, V_h, L_h)$: historical median, CV and first percentile.

**Table 1.** Sampling ranges for multipliers of statistical parameters, where 1 corresponds to the values for the historical time series.

| Sampling Parameter | Lower Bound Multiplier | Upper Bound Multiplier |
|---|---|---|
| Median, $m$ | 0.3 | 1 |
| Coefficient of Variation, $CV$ | 1 | 2 |
| 1st percentile, $q_{low}$ | 0.3 | 1 |

indicate a decrease in the mean discharge values that could reach up to 60% (SYGM, 2016 (accessed December 19, 2022).
An increasing intensity of drought conditions is expected for the period of 2040 - 2071 in the region of the presented case
study (Demircan et al., 2017; Turkes et al., 2020; Yildiz et al., 2022a). In parallel, precipitation variability is widely forecast to
increase (GCMs; Pendergrass et al., 2017), with the coefficient of variation of precipitation projected to almost double by 2060
in various neighboring regions such as the Mediterranean (Giorgi and Lionello, 2008) or Iran (Zarrin and Dadashi-Roudbari,
2021). To reflect these various results while reflecting the uncertainties that surround them, we chose wide ranges for the
scaling factors of our three parameters. These sampling ranges are summarised in Table 1 and reflect the concurrent tendencies
for severe drying and an increase in variability. Recall these ranges represent plausible rather than probable values. We then
sampled $N = 500$ alternative future streamflow conditions using Latin hypercube sampling.

Next, in step (3), we primarily check if the samples satisfy the condition for existence; the smallest and largest measured value of $\frac{CV}{1-R}$ across the sample are 0.75 and 5.15 respectively. All values are significantly larger than the existence condition for the parameterisation ($-1/\ln(\varepsilon) \approx 0.43$, see equation 9). Therefore we can derive the distribution parameters of Kosugi model by using equations (7) to (10) for each sampled future. Thereafter, we generate future scenarios by using these distribution parameters. Figure 3 showcases the versatility of our method and compares to the lack of flexibility provided by a uniform multiplier across the FDC of historical flows. For instance, a uniform 20 % reduction across the flow distribution (dotted black lines) provides a single possible future. For comparison, there are 12 scenarios from our ensemble generated with mean flow reductions ranging from 19 % to 21 % (orange lines), and they display a wide range of low and median flow behaviours, generally lower than the dotted black line, combined with higher high flows. Clearly, our method can provide a suitable range of hydroclimatic conditions, with increased frequency of high flows and low flows, in accordance with likely impacts of climate change in the region. This versatility can be compared to the lack of flexibility offered by a uniform multiplier across the FDC of historical flows, also shown on Figure 3 with the examples of $\pm 20\%$ across the flow distribution (dotted red and black lines).

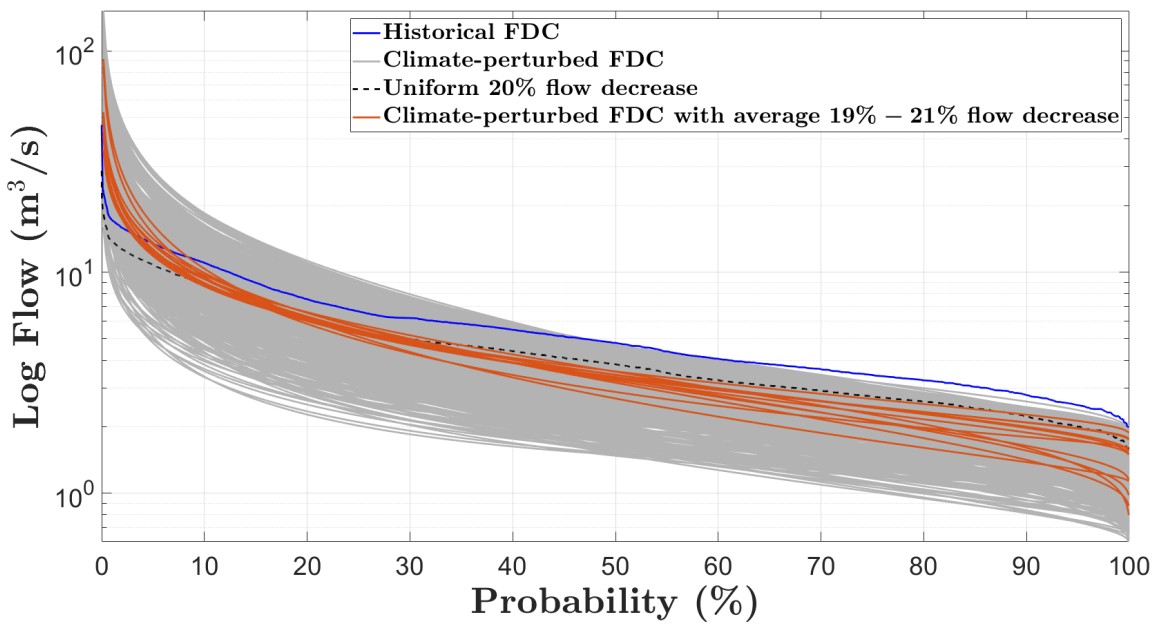

**Figure 3.** Plot of the flow duration curves (FDCs) of the historical record (blue line) and sampled flow duration curves (grey lines) constructed by deriving the FDC parameters for the Kosugi Model shown in Table 1. The figure also compares 20 % mean flow reductions, obtained either with the delta change method (uniform multiplier, dashed black line) and the 12 future scenarios we generated with mean flow reductions between 19 and 21 % (orange lines).

### 3.3 Application to infrastructure robustness

Finally, in step (4), we evaluate the performance of a design under generated future flows. We input each ensemble member into state-of-the-art software to compute technical performance, energy production and economic profit of a design at a given site characteristics (HYPER; Yildiz and Vrugt, 2019). This enables us to quantify the Net Present Value (NPV) of the optimal design of the run-of-river hydropower project under a range of changing climate conditions. The inputs of the HYPER model are daily discharge records, ecological flow requirements, and project-based parameters such as gross head, penstock length,

interest rate, energy price, project life time and site factor for civil works, maintenance and operation cost factor, fixed costs such as transmission line, expropriation costs. Recall that the NPV is the value of projected cash flows, discounted to the present. We assess that the investment is robust to a future climate if NPV is greater than zero. Future FDCs with their respective robustness measure are presented in Figure 4. The Figure shows that although the NPV of current design based on historical records (blue line) is around 10 M$, it decreases dramatically and even becomes negative (gray lines) under dry futures characterised in

particular by a median $m$ under 2.3 $m^3/s$; or a $m$ under 2.6 $m^3/s$ accompanied by $q_{low}$ under 1.10 $m^3/s$ and CV below 0.8. The project is unfeasible under such conditions.

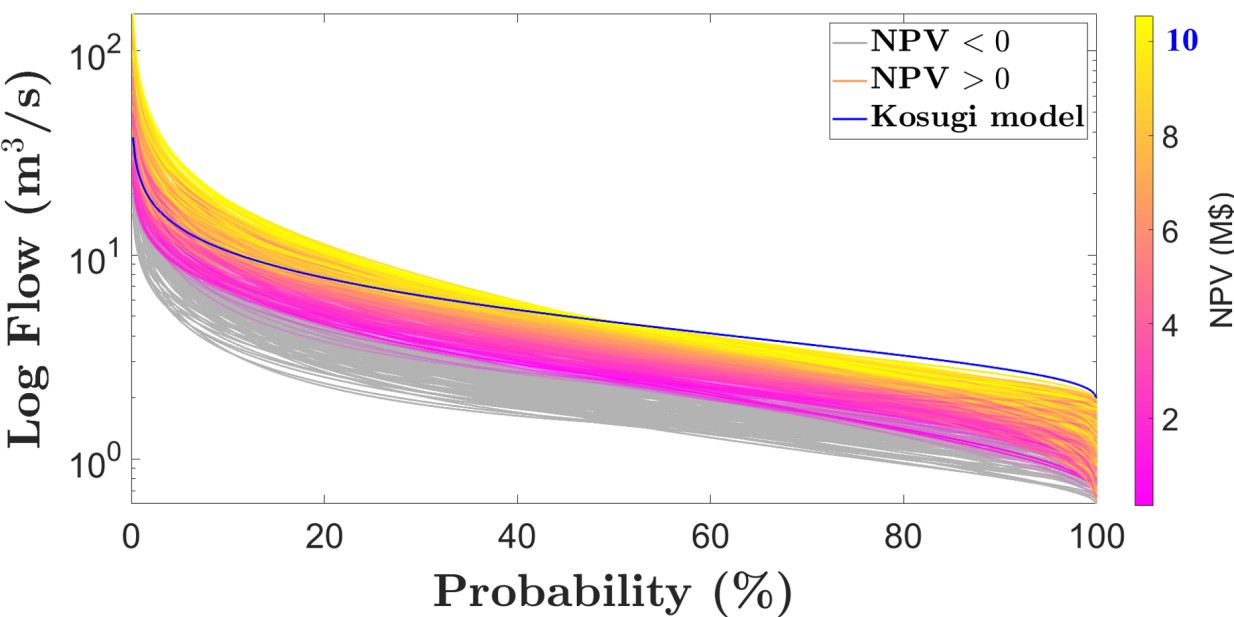

**Figure 4.** Plot of generated flow duration curves (FDCs), with each solution colored by its Net Present Value (NPV). Gray colored lines signifies SOWs in which NPV is negative. NPV of the optimal design based on observed discharge (blue line) is 10 M$

## 4 Discussion and conclusion

In this technical note, we present an effective, practical and novel approach based on a near-universal parameterisation of flow duration curves (FDCs), and perturbation of these parameters to simulate a range of futures in a way that is hydrologically consistent across the spectrum of hydrological conditions. Our application to a run-of-river hydropower project in Eastern Turkey showcases the ability of our method to provide a large range of climate-modified catchment responses, including increased frequency of both high flows and low flows to mimic the future projections for the area (i.e. more arid conditions with increased trend of extreme hydrological events). It compares favorably with existing statistical methods to perturb flows such as the delta change approach. This then supports robustness analyses for rivers for which no detailed hydrological model is available: applied here to assess the financial viability of run-of-river hydropower design in a changing climate. The ease of application of the method illustrates its wide applicability in support of robustness assessments of infrastructure for which streamflow variability impacts performance. We now conclude with some remarks on how this novel approach could be extended to further support such assessments.

Even though the three-parameter Kosugi model has been shown to fit FDCs well across a wide range of catchment characteristics (Sadegh et al., 2016), this does not a guarantee that it would be a good fit in all cases. Sadegh et al. (2016) proposed other functional forms such as the 2-parameter Kosugi model, and 2-parameter and 3-parameter van Genuchten models for the FDC. Despite the superior fit of the 3-parameter Kosugi model across a range of climate zones, these models could also be perturbed to generate future flows.

Our method focuses on catchments free of major flow regulation (reservoir, effluent discharge). Yet, those catchments do not have to be pristine, and can for example experience significant human interference in land use change. Indeed, the MOPEX dataset (Duan et al., 2006), which was used to assess the quality of the three parameter Kosugi model (Sadegh et al., 2016), has been found to be affected by significant human interference (Wang and Hejazi, 2011).

We also identified two current limitations to this method that we believe can be addressed by future developments. First, recent studies reveal that there is an increasing trend of the number of zero-flow days in many regions such as the Mediterranean (e.g., Tramblay et al., 2021). Yet, the number of zero-flow days remains constant in this approach. Preliminary results show that our proposed method supports time series with a large number of zero-flow days, by keeping the number of no-flow days constant and perturbing the FDC when flows are positive. Admittedly, this approximation ignores the fact that a change in climate regime could affect the number of no-flow days. Future work needs to examine the possibility of using the proportion of no-flow days as the low-flow indicator $L$, instead of a low flow quantile. Derivations for the existence and unicity of a parameterisation should then also be extended to that case.

Our approach only considers the FDC, and says nothing of the seasonality, frequency and duration of dry and wet spells. The shifting seasonality of flows in a changing climate can easily be captured by combining our approach with methods such as the log-space rescaling of stationary flows (Quinn et al., 2018) or the reconstruction annual flow hydrographs (Nazemi et al., 2013). Beyond changes in seasonality, there is mounting evidence that climate change is bound to cause hydrological intensification, i.e., it will make dry periods longer and more severe and wet periods more intense (Ficklin et al., 2022).

Information on hydrological intensification scenarios comes from outputs from large-scale climate models, and integrating that information requires turning the climate information into streamflow. One way to do it without the help of a rainfall-runoff model is to control the parameters of a daily streamflow model with a monthly climate model (Stagge and Moglen, 2013). The generation of a FDC for every climate the daily streamflow model simulates could then be used to improve results, e.g., by

providing a quantile-by-quantile adjustment of the synthetic streamflow generator outputs. A similar procedure could combine hydrological model simulations with statistical generation of FDCs. The latter could correct outputs from the former, if they were obtained with a calibration that reflects historical conditions.

*Code and data availability.* The climate-perturbed FDC generation model has been developed in Python 3.10.4. and is provided with an environment file. It is accessible from the Zenodo open-access repository at https://doi.org/10.5281/zenodo.7662679, with a link to the

GitHub source codes of the latest release, including a detailed "run guide" and input files to statistically generate plausible streamflow futures.

*Author contributions.* VY: Conceptualization, Methodology, Writing - original draft, Formal analysis, Software, Investigation. CR: Supervision, Conceptualization, Methodology, Formal analysis, Writing - review & editing, Investigation. RM: Investigation. SB: Conceptualization, Investigation, Supervision. All authors have read and agreed to the published version of the manuscript.

*Competing interests.* The authors declare that they have no known competing financial interests or personal relationships that could have appeared to influence the work reported in this paper.

*Acknowledgements.* Prof. Solomon Brown and Dr Robert Milton are supported by the UK Engineering and Physical Sciences Research Council (EPSRC) through the 'Table Top Manufacturing of Tailored Silica for Personalised Medicine [SiPM]' project (Ref: EP/V051458/1). We also appreciate insights and comments from the Associate Editor Dr Micha Werner as well as two anonymous referees, as they have

greatly improved this manuscript. The first author gratefully acknowledges support from the General Directorate of State Hydraulic Works (DSI-TURKEY).

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
