# Peer review of "Technical note: Statistical generation of climate-perturbed flow duration curves"

_EGUsphere, 2022_

## Author Comment (AC1)

**RESPONSE TO REVIEWER 1 COMMENTS**

Throughout this response, the reviewer's text is presented in black, our response in blue, and the proposed revisions in green. Please also note that line numbers all refer to the current submission.

This technical note introduces a numerical framework for the statistical generation of flow duration curves and then demonstrates its relevance on a hydropower planning problem. The key idea supporting the framework is the representation of Flow Duration Curves (FDC) through a set of parameters, whose value is directly related to key streamflow statistics, e.g., mean, median, or standard deviation. By sampling in the space of these statistics (through the use of multipliers), one can then stochastically generate new FDCs.

I believe the proposed approach is novel and technically sound (including the derivations provided in the SI). Importantly, the proposed approach can indeed be useful for a variety of water management applications. The presentation is clear and the manuscript well structured. Hence, my suggestion is to proceed with a minor revision.

We sincerely appreciate your thoughtful review of our manuscript. Thank you for acknowledging the novelty and technical soundness of our proposed approach, as well as the clarity of the presentation and the manuscript's overall structure.

My only major comment concerns the 'type' of streamflow data that are needed to parameterise the model; a point that, in my opinion, requires a deeper discussion. For example, I believe it may be challenging to implement the framework in a catchment characterized by land use change or other anthropogenic interventions. In other words, I suspect that the use of the framework might be limited to pristine catchments (unless the framework is complemented by a process-based model that somewhat accounts for the aforementioned drivers). Another point I would discuss is the 'safe operating space' of the framework, intended as the amount and quality of data needed for its successful implementation. With this, I am not trying to diminish this paper (which I found interesting), but simply to understand how to best use the model it presents.

Thank you for valuable feedback and insightful suggestions. Below we address separately the two points raised.

We agree with your assessment that our approach is not applicable in any catchment regardless of the amount of human intervention. We will insert the following in the revised manuscript in the discussion, after lines 211-212:

"Our method focuses on catchments free of major flow regulation (reservoir, effluent discharge). Yet, those catchments do not have to be pristine, and can for example experience significant human interference in landuse change. Indeed, the MOPEX dataset (Duan et al., 2006), which was used to assess the quality of the three parameter Kosugi model (Sadegh et al., 2016), has been found to be affected by significant human interference (Wang and Hejazi, 2011). "

Your point regarding the "safe operating space" of the framework is similarly well-made. we will clarify the amount and quality of data required for the successful implementation of our approach after equation (2):

"To fit the Kosugi model and capture flow variability within the FDC, it is necessary to daily discharge measurements over a sufficient period of time, e.g., more than 20 years.".

Finally, the authors may want to consider a full article (rather than a technical note), something that could be done by including the SI in the main manuscript and extending the description of the case study. I would leave this up to the authors.

We would like to thank the reviewer for their suggestion. We carefully considered your suggestion to change the format of this technical note. However, after careful evaluation, we have decided to maintain the current format. Indeed, the supplementary material is there mainly to provide the detailed proof that for any triplet of statistics (M,V,L) there is a unique set of Kosugi parameters; we believe that putting this lengthy proof in the main text would dilute it.

Specific comments

- Abstract: "coherent across the full range of hydrological conditions". Could you please elaborate on or clarify the meaning of this statement?

Thank you. We will amend the text at lines 3-5:

"In this note, we introduce a new statistical generation method to produce a range of plausible streamflow futures to flexibly combine changes in average flows with changes in the frequency and magnitude of high and low flows."

- Line 36-37: I agree with this statement, but also believe that streamflow is not the only source of uncertainty that water planners account for (water demand, for instance, is another one). This is an important caveat I would mention.

We appreciate your suggestion to consider all sources of uncertainty. We will clarify that our paper focuses only on streamflow uncertainty.

For this we will insert at lines 36-38:

"In water resource applications, this entails defining specific ranges for future uncertainties including streamflow, then sampling them to generate an ensemble of plausible future conditions."

- Line 43: should it be "change"?

Thanks for this comment that warrants a clarification.

We will revise this sentence as:

"In fact, a study of the Rhine-Meuse basin from 1901 to 2010 shows that optimal calibration evolves with climate variability, and land use and river structure change (Ruijsch et al., 2021)."

- Line 64. I would say a few words about the Kosugi model. It is hard to follow the next paragraph (and, hence, grasp the overall contribution) without some basic information about the model.

We will add below information to the text at line 65:

"This model is frequently used in soil physics and hydrology to characterise water flow in unsaturated soils and to estimate soil water retention properties. It is based on a lognormal distribution with three parameters (Kosugi, 1994, 1996) that are determined by calibration against the empirical FDC of a watershed."

- Equation 1: I assume that "erfc" refers to the complementary error function, right? I would mention this explicitly in the paper.

We will add the following clarification below the equation

"where [...] erfc is the complementary error function."

- Line 132-133. I'm afraid I don't fully understand this part: why is it necessary to verify this condition?

We appreciate your feedback and the important point you raised regarding the suitability of our model for projecting future outcomes. The ability of the proposed model to fit well with historical observations is critical to its ability to make reliable future projections, and this should be a fundamental consideration.

In a revised version, we will make the need for a good fit model clearer by adding the below text at lines 132-133:

"It is essential to prove that the FDC model provides a good representation of historical observations, otherwise a perturbation of the model would be a poor representation of a perturbation of the historical flow regime."

- Figure 1. I would expand the caption instead of referring the readers to the main text.

We will amend our caption as follows:

Figure 1. Flowchart of the approach; (1) Kosugi model parameters are calibrated with a historical FDC, (2) a set of scenarios with modified flow statistics are determined, and a new set of Kosugi model coefficients are derived for each future scenario. (3) future scenarios are created by using these coefficients, (4) application of the method to represent possible climate change impacts on the robustness of a proposed run of river plant in Turkiye.

- Line 157. "Additional energy"?

Thank you for bringing this to our attention. We will add "additional before "energy" at line 157, to read:

Extreme low flows are insufficient to activate the turbines, and equally, flows above the design discharge do not produce additional energy.

- Line 161. Can you provide more details about the data you used? For instance, how long was this time series? What's the minimum amount of data needed to make the application of this model successful?

In this instance, the information the reviewer is looking for seems to be already present at lines 149 – 150:

"27 years of daily discharge observation are available. The discharge fluctuates considerably between values of 2 and 38 m3/s, with median flow of 4.79 m3/s, first percentile flow of 2.23 m3/s and coefficient of variation of 0.60."

Please also note the 27-year, daily FDC for the catchment is available in the Zenodo repository. We are keen to add more information if reviewers or editors think it is warranted.

Thanks for pointing out our lack of explanation about the minimum amount of data needed to make the application of this model successful. This topic was covered in our earlier justifications.

- Line 189. What are the input variables to HYPER?

Thank you for highlighting the absence of HYPER's input in the text. We will insert this at lines 190:

"The inputs of the HYPER model are daily discharge records, ecological flow rate, and project based parameters such as gross head, penstock length, interest rate, energy selling price, project life time and site factor for civil works, maintenance and operation cost factor, fixed costs such as transmission line, expropriation costs. "

- Line 210. What are these other functional forms?

Thank you for pointing out the need to emphasise this point more explicitly in our paper. We will revise that sentence as:

"Sadegh et al. (2016) proposed other functional forms such as 2-parameter Kosugi model, 2parameter and 3-parameters van Genuchten models for the FDC that could also be perturbed to generate future flows."

References:

Wang, D., & Hejazi, M. (2011). Quantifying the relative contribution of the climate and direct human impacts on mean annual streamflow in the contiguous United States. *Water resources research*, *47*(10).

Kosugi, K. I. (1994). Three-parameter lognormal distribution model for soil water retention. *Water Resources Research*, *30*(4), 891-901.

Kosugi, K. I. (1996). Lognormal distribution model for unsaturated soil hydraulic properties. *Water Resources Research*, *32*(9), 2697-2703.

Thank you again for your thoughtful comments on our manuscript.

---

## Author Response (AR1)

**RESPONSE TO REVIEWER 1 COMMENTS**

**Throughout this response, the reviewer's text is presented in black, our response in blue, and the proposed revisions in green.** Please also note that line numbers all refer to the track-change version.

This technical note introduces a numerical framework for the statistical generation of flow duration curves and then demonstrates its relevance on a hydropower planning problem. The key idea supporting the framework is the representation of Flow Duration Curves (FDC) through a set of parameters, whose value is directly related to key streamflow statistics, e.g., mean, median, or standard deviation. By sampling in the space of these statistics (through the use of multipliers), one can then stochastically generate new FDCs.

I believe the proposed approach is novel and technically sound (including the derivations provided in the SI). Importantly, the proposed approach can indeed be useful for a variety of water management applications. The presentation is clear and the manuscript well structured. Hence, my suggestion is to proceed with a minor revision.

We sincerely appreciate your thoughtful review of our manuscript. Thank you for acknowledging the novelty and technical soundness of our proposed approach, as well as the clarity of the presentation and the manuscript's overall structure.

My only major comment concerns the 'type' of streamflow data that are needed to parameterise the model; a point that, in my opinion, requires a deeper discussion. For example, I believe it may be challenging to implement the framework in a catchment characterized by land use change or other anthropogenic interventions. In other words, I suspect that the use of the framework might be limited to pristine catchments (unless the framework is complemented by a process-based model that somewhat accounts for the aforementioned drivers). Another point I would discuss is the 'safe operating space' of the framework, intended as the amount and quality of data needed for its successful implementation. With this, I am not trying to diminish this paper (which I found interesting), but simply to understand how to best use the model it presents.

Thank you for valuable feedback and insightful suggestions. Below we address separately the two points raised.

We agree with your assessment that our approach is not applicable in any catchment regardless of the amount of human intervention. We inserted the following in the revised manuscript in the discussion, after lines 249-252:

"Our method focuses on catchments free of major flow regulation (reservoir, effluent discharge). Yet, those catchments do not have to be pristine, and can for example experience significant human interference in land use change. Indeed, the MOPEX dataset (Duan et al., 2006), which was used to assess the quality of the three parameter Kosugi model (Sadegh et al., 2016), has been found to be affected by significant human interference (Wang and Hejazi, 2011). "

Your point regarding the "safe operating space" of the framework is similarly well-made. we clarified the amount and quality of data required for the successful implementation of our approach after equation (2):

"To fit the Kosugi model and capture flow variability within the FDC, it is necessary to daily discharge measurements over a sufficient period of time, e.g., more than 20 years.".

Finally, the authors may want to consider a full article (rather than a technical note), something that could be done by including the SI in the main manuscript and extending the description of the case study. I would leave this up to the authors.

We would like to thank the reviewer for their suggestion. We carefully considered your suggestion to change the format of this technical note. However, after careful evaluation, we have decided to maintain the current format. Indeed, the supplementary material is there mainly to provide the detailed proof that for any triplet of statistics (M,V,L) there is a unique set of Kosugi parameters; we believe that putting this lengthy proof in the main text would dilute it.

Specific comments

- Abstract: "coherent across the full range of hydrological conditions". Could you please elaborate on or clarify the meaning of this statement?

Thank you. We amended the text at lines 5-7:

"In this note, we introduce a new method for the statistical generation of plausible streamflow futures. It flexibly combines changes in average flows with changes in the frequency and magnitude of high and low flows."

 - Line 36-37: I agree with this statement, but also believe that streamflow is not the only source of uncertainty that water planners account for (water demand, for instance, is another one). This is an important caveat I would mention.

We appreciate your suggestion to consider all sources of uncertainty. We will clarify that our paper focuses only on streamflow uncertainty.

For this we  inserted at lines 37-39:

"In water resource applications, this entails defining specific ranges for future uncertainties including streamflow, then sampling them to generate an ensemble of plausible future conditions."

- Line 43: should it be "change"?

Thanks for this comment that warrants a clarification.

We revised this sentence at lines 44-45 as:

"In fact, a study of the Rhine-Meuse basin from 1901 to 2010 shows that optimal calibration evolves with climate variability, and land use and river structure change (Ruijsch et al., 2021)."

- Line 64. I would say a few words about the Kosugi model. It is hard to follow the next paragraph (and, hence, grasp the overall contribution) without some basic information about the model.

We added below information to the text at lines 63-72:

"This remark has led Sadegh et al. (2016) to adapt a set of soil retention functions such as those proposed by van Genuchten (1980) and Kosugi (1996) to mimic the empirical FDCs of catchments. These models are used in soil physics and hydrology to characterise water flow in unsaturated soils and to estimate soil water retention properties. This analogy is based on the idea that both watersheds and soils are governed by similar hydroclimatologic forcing, and are able to store and dispel precipitation in response to similar gradients (Vrugt and Sadegh 2013; Sadegh et al., 2016). Fitting FDCs to a set of 430 catchments of the MOPEX dataset (Duan et al., 2006), Sadegh et al. (2016) found that the three-parameter Kosugi model they proposed offered the best quality of fit across a broad range of climate zones, under a goodness-of-fit criterion that weighs high and low flows equally. It is based on a lognormal distribution with three parameters (Kosugi, 1994, 1996) that are determined by calibration against the empirical FDC of a watershed."

- Equation 1: I assume that "erfc" refers to the complementary error function, right? I would mention this explicitly in the paper.

We added the following clarification below the equation at line 91:

"where [...], and erfc is the complementary error function."

- Line 132-133. I'm afraid I don't fully understand this part: why is it necessary to verify this condition?

We appreciate your feedback and the important point you raised regarding the suitability of our model for projecting future outcomes. The ability of the proposed model to fit well with historical observations is critical to its ability to make reliable future projections, and this should be a fundamental consideration.

In a revised version, we made the need for a good fit model clearer by adding the below text at lines 158-160:

"It is essential to prove that the FDC model provides a good representation of historical observations, otherwise a perturbation of the model would be a poor representation of a perturbation of the historical flow regime."

- Figure 1. I would expand the caption instead of referring the readers to the main text.

We amended our caption as follows:

Figure 1. Flowchart of the approach; (1) Kosugi model parameters are calibrated with a historical FDC, (2) a set of scenarios with modified flow statistics are determined, (3) a new set of Kosugi model coefficents are derived for each future scenario, and future scenarios are

created by using these coefficients, (4) future scenarios can be used in robustness assessments.

- Line 157. "Additional energy"?

Thank you for bringing this to our attention. We added "additional before "energy" at line 184, to read:

"Extreme low flows are insufficient to activate the turbines, and equally, flows above the design discharge do not produce additional energy."

- Line 161. Can you provide more details about the data you used? For instance, how long was this time series? What's the minimum amount of data needed to make the application of this model successful?

In this instance, the information the reviewer is looking for seems to be already present at lines 176 – 177:

"27 years of daily discharge observation are available. The discharge fluctuates considerably between values of 2 and 38 $m^3$/s, with median flow of 4.79 $m^3$/s, first percentile flow of 2.23 $m^3$/s and coefficient of variation of 0.60."

Please also note the 27-year, daily FDC for the catchment is available in the Zenodo repository. We are keen to add more information if reviewers or editors think it is warranted.

Thanks for pointing out our lack of explanation about the minimum amount of data needed to make the application of this model successful. This topic was covered in our earlier justifications.

- Line 189. What are the input variables to HYPER?

Thank you for highlighting the absence of HYPER's input in the text. We inserted this at lines 222-225:

"The inputs of the HYPER model are daily discharge records, ecological flow requirements, and project-based parameters such as gross head, penstock length, interest rate, energy price, project life time and site factor for civil works, maintenance and operation cost factor, fixed costs such as transmission line, expropriation costs. "

- Line 210. What are these other functional forms?

Thank you for pointing out the need to emphasise this point more explicitly in our paper. We revised that sentence at lines 244-247 as:

"Sadegh et al. (2016) proposed other functional forms such as the 2-parameter Kosugi model, and 2-parameter and 3-parameter van Genuchten models for the FDC. Despite the superior fit of the 3-parameter Kosugi model across a range of climate zones, these models could also be perturbed to generate future flows."

References:

Wang, D., & Hejazi, M. (2011). Quantifying the relative contribution of the climate and direct human impacts on mean annual streamflow in the contiguous United States. *Water resources research*, *47*(10).

Kosugi, K. I. (1994). Three- parameter lognormal distribution model for soil water retention. *Water Resources Research*, *30*(4), 891-901.

Kosugi, K. I. (1996). Lognormal distribution model for unsaturated soil hydraulic properties. *Water Resources Research*, *32*(9), 2697-2703.

Thank you again for your thoughtful comments on our manuscript.

**RESPONSE TO REVIEWER 2 COMMENTS**

**Throughout this response, the reviewer's text is presented in black, our response in blue, and the proposed revisions in green.** Please also note that line numbers all refer to the track-change version.

Yildiz et al. introduce a new approach to generate possible future streamflow scenarios for stress testing the impacts of possible climatic changes on river systems. The approach is elegant, requiring only three parameters to modify key characteristics of the flow duration curve (mean, standard deviation and low/high flow quantile or median, coefficient of variation and low/high flow quantile). I think this approach is a nice contribution to the literature. I have a only a few suggestions for improvement.

We sincerely appreciate your thoughtful review of our work. We are happy to hear that you consider our contribution to be a valuable addition to the existing literature. Furthermore, we are grateful for your suggestions, which we believe will strengthen this paper.

The Discussion claims that this method "compares favorably with existing statistical methods to perturb flows such as the delta change approach." However, the paper does not formally compare the proposed FDC alteration approach with the delta change approach. I think it would help sell the method to include a few FDC alterations with the same mean change but different changes in the variance and low flow quantile with using the delta change method to achieve the same mean change. Seeing differences in both the streamflow time series and resulting performance impacts from the delta change method vs. different FDC alterations that achieve the same mean change would help sell the utility of this approach for climate vulnerability assessments.

We appreciate your insightful input. Based on your suggestion, we recognize the importance of providing a direct comparison between the two methods in a visual manner.

To address this, we revised Figure 3 as follows below, to clearly illustrate the differences between our proposed method and the delta approach. This addition enhanced the clarity and comprehensiveness of our study.

[Figure]

Figure 3. Plot of the flow duration curves (FDCs) of the historical record (blue line) and sampled flow duration curves (grey lines) constructed by deriving the FDC parameters for the Kosugi Model shown in Table 1. The figure also compares 20% mean flow reductions, obtained either with the delta change method (uniform multiplier, dashed black line) and future scenarios we generated with mean flow reductions between 19.5 and 20.5% (orange lines).

We also amended the accompanying text accordingly at lines 208-213

"Figure 3 showcases the versatility of our method and compares to the lack of flexibility provided by a uniform multiplier across the FDC of historical flows. For instance, a uniform 20% reduction across the flow distribution (dotted black lines) provides a single possible future, whereas scenarios from our ensemble generated with comparable mean flow reductions – ranging from 19.5% to 20.5% (orange lines) – display a wide range of low and median flow behaviours, generally lower than the dotted black line, combined with higher high flows."

Discuss the conditions of unicity (either when they are introduced or in the Discussion section). Are these conditions likely to be met, and if so why? Where might it not be true? What are the implications of not being able to explore changes that don't meet these conditions?

Thank you for highlighting the importance of existence conditions in our approach. We agree with these conditions being crucial, and we verify them across our ensemble in our application (lines 201-203). In general, we believe that this condition must be verified on a case by case basis across the generated ensembles of future flows. We also believe a general validation based on historical flows is of limited relevance, because the condition needs to be verified for all future flows in the generated ensemble, and not just for historical flows.

This being said, we are happy to give hints as to why this condition will be valid most of the time. Indeed, equations (5) and (9) are both equivalent to CV > -(1-R)/ ln(ε) where 0<R<1 is a ratio of the low flows by the mean or median.

Therefore, a sufficient condition for a unique solution to exist is that the coefficient of variation CV > -1/ ln(ε), which corresponds approximately to CV > 0.43 if the low flow indicator is the

first percentile, and CV > 0.61 if it is the fifth percentile. This sufficient condition has been verified for most catchments over a wide dataset of 6807 gages in the continental US (see Ye et al., 2021). And when this sufficient condition is not met, one also needs 1-R to be close to one for the existence condition to be violated. In other words, one needs low flows to be very low in comparison to the mean (for the "mean" case) or median (for the "median" case) and this is a condition that tends to increase the value of CV. In fact, in Ye et al. (2021) figure 10, all time series with zero flow days have a CV value close or equal to 1.

To clarify this in the text, we added a separate section 2.2.3 to comment on the conditions of equation (5) and (9):

2.2.3 Domain of validity of existence conditions

In this paragraph, we explain what the conditions for the existence and uniqueness provided imply – see equations (5) and (9) for "mean" case and "median" case respectively. Both equations are equivalent to:

$$CV > \frac{-(1-R)}{\ln(\varepsilon)}$$

(11)

where 0<R<1 is a ratio of the low flows by the mean or median; recall that $-1/\ln(\varepsilon) \approx 0.43$ if the low flow parameter is the first percentile, or 0.61 if it is the fifth percentile.

From equation (11), it is sufficient to have CV > -1/ ln($\varepsilon$) for both existence conditions to be verified. This condition has been verified for a large majority of the catchments over a large dataset of 6807 gages in the continental US (see Ye et al., 2021). Yet for the existence condition to not be met the multiplier of (1-R) must also be close to 1. In other words, low flows must be extremely low relative to the mean (for the "mean" case) or median (for the "median" case), but this may be incompatible with a low value of CV. In fact, in Figure 10 from Ye et al. (2021), all time series with zero flow days in the sample have a CV value close or equal to 1. Together, these remarks suggest that the existence condition should be realised in most cases where flows are not strongly regulated. However, we would like to point out that whether the conditions of equations (5) or (9) are met for historical flows is of limited relevance. They need to be verified for each plausible future flow for which a FDC is generated. For this reason, we consider that checking these conditions across large databases of historical flows would be of limited interest within the scope of this work.

One noted limitation in the Discussion of this FDC alteration is it does not change the length of wet and dry spells. I recommend noting this can be achieved by changing the parameters of a Markov chain-based streamflow generator (see e.g. Stagge and Moglen, 2013).

Another limitation of the FDC approach not mentioned in the Discussion is that it cannot capture changes in seasonality, which would preclude its application in snow-dominated catchments, or perhaps monsoon systems. I recommend noting this as well. See examples in the literature from Nazemi et al. (2013) and Quinn et al. (2018).

We appreciate your input on proposing alternative methods to address the limitations of our approach. We will address the two comments together. We think that suggested methods

could be a possible solution to address aforementioned limitations. We added below text in the discussion section of the manuscript, by amending its last paragraph as follows:

"Our approach only considers the FDC, and says nothing of the seasonality, frequency and duration of dry and wet spells. The shifting seasonality of flows in a changing climate can easily be captured by combining our approach with methods such as the log-space rescaling of stationary flows (Quinn et al. 2018) or the reconstruction annual flow hydrographs (Nazemi et al. 2013). Beyond changes in seasonality, there is mounting evidence that climate change is bound to cause hydrological intensification, i.e., it will make dry periods longer and more severe and wet periods more intense (Ficklin et al., 2022). Information on hydrological intensification scenarios comes from outputs from large-scale climate models, and integrating that information requires turning the climate information into streamflow. One way to do it without the help of a rainfall-runoff model is to control the parameters of a daily streamflow model with a monthly climate model (Stagge and Moglen 2013). The generation of a FDC for every climate the daily streamflow model simulates could then be used to improve results, e.g., by providing a quantile-by-quantile adjustment of the synthetic streamflow generator outputs. A similar procedure could combine hydrological model simulations with statistical generation of FDCs. The latter could correct outputs from the former, if they were obtained with a calibration that reflects historical conditions."

Minor comments:

Line 70: drop "of" after "represent"

Line 140: change "Zenedo" to "Zenodo"

Line 159: change "standard deviation" to "coefficient of variation"

Line 171: drop "is" before "projected"

Line 176: change "latin" to "Latin"

Line 177: "the" is repeated

Thank you for bringing the typing errors to our attention. We revised the manuscript to correct all the identified typing errors.

Table 1: why not explore potential increases in the median/1st percentile or decreases in the coefficient of variation?

Thanks for this comment. The reviewer is perfectly right that the method can be applied to explore opposite changes to those described in Table 1. Yet, our illustration of our methodology focuses on a region where all studies point to a drier and more variable future, as justified in lines 193-199. This explains the parameter ranges chosen in Table 1.

References:

Nazemi, A., Wheater, H. S., Chun, K. P., & Elshorbagy, A. (2013). A stochastic reconstruction framework for analysis of water resource system vulnerability to climate-induced changes in river flow regime. Water Resources Research, 49(1), 291-305.

Quinn, J. D., Reed, P. M., Giuliani, M., Castelletti, A., Oyler, J. W., & Nicholas, R. E. (2018). Exploring how changing monsoonal dynamics and human pressures challenge multireservoir management for flood protection, hydropower production, and agricultural water supply. Water Resources Research, 54(7), 4638-4662.

Stagge, J. H., & Moglen, G. E. (2013). A nonparametric stochastic method for generating daily climate-adjusted streamflows. Water Resources Research, 49(10), 6179-6193.

Ye, L., Gu, X., Wang, D., & Vogel, R. M. (2021). An unbiased estimator of coefficient of variation of streamflow. *Journal of Hydrology*, *594*, 125954.

Thank you again for your thoughtful comments on our manuscript.

---

## Author Response (AR2)

**RESPONSE TO EDITOR COMMENTS**

Throughout this response, the editor's text is presented in black, our response in blue, and the proposed revisions in green. Please also note that line numbers all refer to the revised version.

1.In my previous response I noted some minor grammatical errors in the responses provided. Please check these responses again as the errors (including the one explicitly mentioned) are still in the revised manuscript (see e.g. line 94/95).

Thank you for bringing the typing errors to our attention. We revised the manuscript to correct all the identified typing errors. In particular lines 94-95:

"To fit the Kosugi model and capture flow variability within the FDC, it is necessary to have daily discharge measurements over a sufficient period of time, e.g., more than 20 years."

Please let us know if you identify any outstanding typing errors and we'll be happy to correct them.

2. In response you also include in Fig 3 a climate perturbation of the FDC with a range of 19.5% - 20.5%. I understand that this is in the same range as the 20% decrease suggested by the Delta method, but why is the range so small? Would a larger range not approximate the FDC's sampled through the fitted Kosugi model?

We appreciate you bringing up the importance of providing an explanation for our choice of the range in Figure 3. Based on your suggestion we increased the range to 19 % - 21 % which is large enough to yield 12 different FDCs featuring a range of behaviours, demonstrating the added flexibility of our method.

Figure 3. Plot of the flow duration curves (FDCs) of the historical record (blue line) and sampled flow duration curves (grey lines) constructed by deriving the FDC parameters for the

Kosugi Model shown in Table 1. The figure also compares 20 % mean flow reductions, obtained either with the delta change method (uniform multiplier, dashed black line) and the 12 future scenarios we generated with mean flow reductions between 19 and 21 % (orange lines).

We also amended the accompanying text accordingly at lines 206-209

"Figure 3 showcases the versatility of our method and compares to the lack of flexibility provided by a uniform multiplier across the FDC of historical flows. For instance, a uniform 20 % reduction across the flow distribution (dotted black lines) provides a single possible future. For comparison, there are 12 scenarios from our ensemble generated with mean flow reductions ranging from 19 % to 21 % (orange lines), and they display a wide range of low and median flow behaviours, generally lower than the dotted black line, combined with higher high flows."

3. Though not a big issue, in fig 1 you show the workflow. The last step of this shows a figure to assess the robustness of the design. However, the figure you use to assess the robustness (fig. 4) follows the design of the FDC figures. That appears somewhat strange. Would it now make sense to include an inset of fig 4 in fig 1.

We appreciate your insightful input. Based on your suggestion we revised the last step of Figure 1

Please note that we also amended the Acknowledgements section.

Thank you again for your thoughtful comments on our manuscript.